# Redox and Thiols in Archaea

**DOI:** 10.3390/antiox9050381

**Published:** 2020-05-05

**Authors:** Mamta Rawat, Julie A. Maupin-Furlow

**Affiliations:** 1Biology Department, California State University, Fresno, CA 93740, USA; 2Department of Microbiology and Cell Science, Institute of Food and Agricultural Sciences, University of Florida, Gainesville, FL 32611, USA; 3Genetics Institute, University of Florida, Gainesville, FL 32611, USA

**Keywords:** archaea, low molecular weight thiols, glutathione, γ-glutamylcysteine, coenzyme A, redox cycling

## Abstract

Low molecular weight (LMW) thiols have many functions in bacteria and eukarya, ranging from redox homeostasis to acting as cofactors in numerous reactions, including detoxification of xenobiotic compounds. The LMW thiol, glutathione (GSH), is found in eukaryotes and many species of bacteria. Analogues of GSH include the structurally different LMW thiols: bacillithiol, mycothiol, ergothioneine, and coenzyme A. Many advances have been made in understanding the diverse and multiple functions of GSH and GSH analogues in bacteria but much less is known about distribution and functions of GSH and its analogues in archaea, which constitute the third domain of life, occupying many niches, including those in extreme environments. Archaea are able to use many energy sources and have many unique metabolic reactions and as a result are major contributors to geochemical cycles. As LMW thiols are major players in cells, this review explores the distribution of thiols and their biochemistry in archaea.

## 1. Introduction

Low molecular weight (LMW) thiols, a group of highly reactive compounds containing a sulfhydryl (–SH) functional group, play critical roles within a cell [1]. Thiols can: (i) serve as a storage form for cysteine, which can rapidly auto-oxidize in the presence of metals especially copper and iron [2,3], (ii) directly donate electrons to oxidants becoming oxidized in the process, forming disulfides (RS-SR) [4,5], (iii) conjugate xenobiotic agents making them more soluble [6], (iv) form complexes with metal ions [7], and (v) act as cofactors to different enzymes, such as ribonucleotide reductases [8] and methionine sulfoxide reductases [9]. Because of these roles, LMW thiols are considered ubiquitous among living organisms.

Glutathione (GSH) is a tripeptide of γ-l-glutamyl-l-cysteinylglycine that represents the major LMW thiol among eukaryotes, whereas prokaryotes more commonly synthesize alternative LMW thiols even if GSH is present [1]. Ergothioneine (EGT) [10,11,12], trypanathione, bis-glutathione [13], glutathione amide [14], γ-glutamylcysteine (γGC), mycothiol (MSH) [15,16], coenzyme A (CoA), and bacilithiol (BSH) are some of the other LMW thiols (Figure 1). Interestingly, even in bacteria that contain GSH, this LMW thiol can be acquired by pathways that differ from the canonical pathway in which GSH is synthesized by the two ligases, γ-glutamate-cysteine ligase (γ-ECL) and glutathione synthetase (GS). These alternative pathways of GSH acquisition include import [17,18] and the use of a novel fusion protein, GshF, for the biosynthesis of GSH in bacteria that were once thought to lack GSH [19]. The diversity in structure of LMW thiols is now being appreciated and their functions are slowly being unraveled.

In recent years, much progress has been made in elucidating the structure and role of alternative LMW thiols in bacteria. *S*-thiolation of proteins in organisms containing alternative LMW thiols has been demonstrated [20]. However, the distribution, structure, and function of LMW thiols in archaea is still not apparent. This review paper will provide an overview of LMW thiols in prokaryotes with emphasis on the current state of knowledge regarding the distribution and biochemistry of these thiols in archaea.

## 2. Glutathione and γ-Glutamylcysteine

Life on earth started in a reducing environment but the introduction of oxygen demanded an antioxidant network to cope with the oxidizing conditions. The dogma in the field has been that the antioxidant network consisted of GSH, as this LMW thiol is common in eukaryotes [21] and Gram-negative bacteria [16,22]. Thus, much attention has been devoted to finding this LMW thiol in archaea. The majority of bacterial and archaeal phyla have anaerobic basal members with aerobic members found only in derived positions. However, even in these anaerobic members, antioxidant networks are needed (e.g., Bacteriodes) and may have co-evolved as the oxygen levels rose [23].

GSH is synthesized in two sequential enzymatic steps [24]. The *gshA*-encoded γGC synthetase or ligase (GshA, EC 6.3.2.2; γ-ECL) ligates the amino group of cysteine to the γ-carboxyl group of glutamate. In turn, the *gshB*-encoded glutathione synthetase (GshB, EC 6.3.2.3; GS) condenses the resulting γGC with glycine to generate GSH. A bifunctional GshF (GshAB) also exists that catalyzes both steps of GSH synthesis; the encoding gene appears to have spread by horizontal transfer in bacterial symbionts or pathogens, since most of the bacteria containing *gshF* gene homologs are found in domestic animals or humans [25]. Modifications of GSH exist as exemplified by the glutathione amide present in the purple green *Chromatium* [14,26], and ovothiol [27], and trypanothione [28] found mainly in unicellular eukaryotic parasites.

The first comprehensive report of a LMW thiol and its function in archaea was of γGC in the halophilic archaeon (haloarchaeon) *Halobacterium salinarum* (*Halobacterium halobium* R1) by Newton and Javor in 1985 [29]. Other haloarchaea (i.e., *Haloarcula californiae*, *Haloarcula* (*Halobacterium*) *marismortui*, *Halobacterium saccharovorum*, *Haloferax* (*Halobacterium*) *volcanii*, and *Halococcus* sp. LS-1) were also found to contain γGC [29]. Haloarchaea live in high salt environments, greater than 3M NaCl, and possess cytoplasms with ionic strengths similar or exceeding those of their environment [30]. Sundquist and Fahey (1989) [31] demonstrated that the auto-oxidation of γGC in the presence of copper was substantially lower in high salt buffer than low salt buffer and that γGC was more stable in the high salt buffer than GSH was in the low salt buffer. Further analysis revealed the cytosolic concentration of γGC was 4 mM, 50 fold higher than the oxidized γGC (i.e., bis-γGC) in haloarchaea such as *H. salinarum* (*halobium*) [31]. While the gene was not identified in this early work, Sundquist and Fahey provided evidence in *H. salinarum* for a γGC reductase (GCR) that was distinct from dihydrolipoamide dehydrogenase (DHD), the E3 component that oxidizes the thiol groups of dihydrolipoamide (Lip-(SH)_2_) to lipoamide (Lip-(S)_2_) in α-keto acid dehydrogenase complexes [31,32]. Kim and Copley (2013) [33] further expressed an *H. salinarum* (sp. NRC-1) gene annotated as mercuric reductase (MerA) in *Escherichia coli* and demonstrated that the enzyme had robust NADPH-dependent GCR activity but no mercuric reductase activity [33]. The genomes of most, but not all, haloarchaea for which whole genome sequences are available have homologues that are at least 50% identical to GCR (UniRef 50 of UniProt Q9HSN0). These homologs, while uncharacterized, are often annotated as DHDs. However, haloarchaea use ferredoxin-dependent oxidoreductases to oxidize α-keto acids such as pyruvate and α-ketoglutarate, and, thus, these E3 homologs may not be needed for central metabolism [34,35] and instead may code for GCRs. Malki et al. (2009) [36] further showed that *H. volcanii*
*gshA* (HVO_1668) is able to synthesize γGC in vivo. This haloarchaeal *gshA* gene can also restore synthesis of GSH in an *E. coli*
*gshA* mutant despite only 15% sequence identity [36]. The phylogenetic analysis of the *H. volcanii* GshA demonstrated that it clusters with at least 10 other haloarchaeal GshA homologs at 64–75% identity [36]. Thus, GshA and GCR are likely used to synthesize γGC and reduce the oxidized form of this LMW thiol in haloarchaea, respectively.

Genome mining suggests that, in addition to haloarchaea, other archaea may synthesize γGC and/or GSH. In the early survey of LMW thiols in archaea, Newton and Javor (1985) reported that *Sulfolobus acidocaldarius* and five unidentified methanogens did not contain GSH or γGC [29]. However, a putative GshA (Msp_0528) from the methanogen *Methanosphaera stadtmanae*, was identified to form a distinct cluster that was related to haloarchaeal GshA, standalone γ-ECLs, and the N-terminal γ-ECL domain of bifunctional GshFs [36]. This GshA homolog is conserved in other *Methanosphaera* species and has over 50% sequence identity to homologs of *Methanobrevibacter* species including *Methanobrevibacter ruminantium* M1, which is responsible for the ruminant methane in ruminants worldwide [37]. While these latter ORFs are sometimes annotated as bifunctional GshFs, only the N-terminal γ-ECL domain is conserved suggesting γGC ligase, but not GS activity.

The InterPro database has two family classifications for GshA homologs, IPR006334 and IPR006336, which account for 676 total hits in archaea (Figure 2; Appendix A). Phylogenetic analysis by Copley and Dhillon (2002) indicates that GshA sequences fall into three groups including those primarily from: (i) γ-proteobacteria, (ii) non-plant eukaryotes, and (iii) α-proteobacteria and plants, with the latter including sequences from haloarchaea [38]. From this early analysis, γ-ECL genes were suggested to have originated in cyanobacteria then to have undergone horizontal gene transfer (HGT) to other bacteria, eukaryotes, and at least some archaea [38]. Of the archaeal hits to the GshA InterPro families, most (496/676; 73%) are classified to the haloarchaea (*Halobacteria* class). The other hits are dispersed among other Euryarchaeota (methanogens, *Archaeoglobi*, and *Thermoplasmata*) and representatives of the Asgard, DPANN and TACK superphyla (Figure 2). While no archaeal hits for GshB (IPR037013) or GshF (GshAB) (IPR006335) are apparent, some H_2_-oxidizing methanogens (*Methanobacterium* sp.) have standalone GshB-like proteins related to the C-terminal ATP-grasp-like domain (IPR040657) of the GshAB fusion GshF. In genome synteny with these GshB-like ORFs are genes encoding GshA-like proteins related to the N-terminal domain of GshF (GshAB) suggesting certain methanogens may synthesize GSH.

## 3. Protein Disulfide Relay Systems

Small protein-based dithiols can serve directly or indirectly in disulfide relays as reductants for enzymatic reactions as well as for the repair or regulation of proteins that undergo oxidation [40]. Examples include the reduction of disulfide bonds created on enzymes as part of the formation of deoxyribonucleotides for DNA synthesis by reducing ribonucleotide reductase [8], the generation of reduced sulfur via 3’-phosphoadenylylsulfate [41], the reduction of methionine sulfoxide (an oxidized form of methionine) in the repair or regulation of oxidized proteins [9], the detoxification of H_2_O_2_ [42] and other activities of the cell. Most notable among the protein-based dithiol reductants are the thioredoxins (TRXs) and glutaredoxins (GRXs) that use active site cysteine residues in a CXXC motif. These cysteine residues are oxidized to form a disulfide bond upon transfer of the reductant to the substrate or enzyme and require regeneration by reduction for future reactions to proceed (Figure 3) [43,44]. For GRXs, the disulfide bond formed in the active site after reductant transfer can be reduced by LMW thiols, which in turn are oxidized. For example two GSHs can reduce the GRX disulfide resulting in GSH oxidation to GSSG, which is reduced by GSH reductase (EC 1.8.1.7), using reducing equivalents from NADPH [45]. By contrast to GRXs, TRXs are reduced enzymatically by TRX reductases which can be classified by active site and electron donor. Included in this classification are the: (i) NTRs, NAD(P)H-dependent TRX reductase flavoproteins (contain FAD coenzyme; EC 1.8.1.9) [46], with some also able to use hydrogen (H_2_) [47], (ii) DFTRs, deazaflavin (F_420_)-dependent TRX reductase flavoproteins [48], (iii) FFTRs, ferredoxin (Fd)-dependent TRX reductase flavoproteins of certain bacteria [49], and iv) FTRs/FDRs, Fd: TRX reductases and Fd: disulfide reductases that use an active-site [4Fe–4S] cluster [50] (Figure 3). The F_420_ cofactor used by the DFTRs is distinct from FAD and is common to methanogenic archaea (Figure 4).

Thiol transferase systems that use protein dithiol reductants and TRX reductases are reported in archaea, with some of the archaeal TRXs characterized so far having TRX activity but an apparent GRX-like structure. NTRs that bind FAD and reduce TRXs using NAD(P)H are observed in the hyperthermophilic crenarchaeota *Saccharolobus* (*Sulfolobus*) *solfataricus* [51,52] and *Aeropyrum pernix* K1 [53]. NTRs are also reported in hyperthermophilic euryarchaeota, such as *Pyrococcus horikoshii* [54] and *Thermococcus onnurineus,* with the latter incorporating a TRX reductase that can also directly use H_2_ as reductant [47]. A GRX-like protein from *P. horikoshii* has demonstrated TRX activity but no GRX activity, meaning that it requires a TRX reductase for reactivation [54]. Likewise, a GRX-like protein disulfide oxidoreductase (PDO), is described in *S. solfataricus* to accept reductant from NTR and transfer this to enzymatic reactions in the cell [52,55]. Most methanogens have multiple TRX homologues with distinct functions [56]. One exception is *Methanopyrus kandleri* of the order *Methanopyrales*, which has a complete genome sequence and no predicted TRX homolog [57]. Constant with this TRX distribution in methanogens, NTR and FDR enzymes are observed in *Methanosarcina acetivorans* [50,56]. Likewise, an FFTR was identified in the hyperthermophilic methanogen *Methanocaldococcus jannaschii*, that can reduce TRXs [48] as well as a protein similar in structure to GRX that acts like a TRX [58]. *Methanobacterium thermoautotrophicum,* a methanogen that grows optimally at ~65 °C, contains a GRX-like protein [59]. GSH, TRX or the two thiols identified in extracts of *M. thermoautotrophicum,* hydrogen sulfide and 2-mercaptoethanesulfonate (coenzyme M, CoM), were not able to serve as reductants but dihydrolipoate could in a GRX-like activity assay that monitored the reduction of insulin disulfide [59]. In addition, a GRX-like protein in *M. acetivorans*, was named methanoredoxin (MRX) [60] because it used CoM-SH as reductant in the insulin disulfide reductase assay. CoM along with coenzyme B (CoB) are thiol-based coenzymes that form a heterodisulfide (CoM-S-S-CoB) during methanogenesis [61]. Consistent with the possibility that CoM-SH could serve as a GRX-like reductant, a CoM disulfide reductase is described in *M. thermoautotrophicum* that regenerates CoM-S-S-CoM to CoM-SH [62]. In γGC containing haloarchaea, thioltransferase activity is observed and suggested to be used to reduce disulfides which are part of an enzymatic cycle [63]; alternatively, in the absence of a LMW thiol, TRX could perform the functions attributed to the GRX-like activity [64]. InterPro search for GRX-like homologs in archaea yielded 2892 hits, while the TRX-like fold yielded another 2775 hits (Appendix A) with many of these proteins having the conserved CXXC motif suggesting thiol-disulfide oxidoreductase activity is common in archaea.

## 4. Thiol-Dependent Peroxidases

One of the critical functions of LMW thiols is the reduction of oxidants. In eukaryotes, GSH peroxidases, GPX(s), are able to detoxify organic hydroperoxides, such as lipid peroxides [65]. The general absence of polyunsaturated fatty acids in archaea and bacteria suggests that GSH-dependent thiol peroxidases evolved in eukaryotes in response to the need to protect against polyunsaturated fatty acid oxidation. Assays for GSH and γGC-dependent enzymes demonstrated an absence of peroxidase activity in bacteria and haloarchaea [63]. *E. coli* BtuE is the only bacterial enzyme sharing homology with the GPX family that has been shown to have GSH dependent peroxidase activity with H_2_O_2_; however, BtuE is able to use thioredoxin (TRX, see below) to reduce organic peroxides and prefers TRX over GSH for the reduction of H_2_O_2_ [66]. In addition, γGC could serve as a cofactor for human GPx1-mediated H_2_O_2_ reduction with similar efficiency as GSH only at low concentrations of the thiols [67]. InterPro (IPR000889) search of GSH peroxidases yielded 102 archaeal hits (Figure 2) suggesting that either a GSH or γGC dependent peroxidase may exist in archaea. Peroxiredoxins (Prxs), thiol-dependent peroxidases that scavenge peroxides, have been identified in archaea [68,69,70,71,72,73,74,75]. In particular, the *S. solfataricus* Prx is shown to use a thiol relay system in which the GRX-like PDO accepts reductant from NTR to reactivate the Prx in its catalytic cycle [68,69,70,71,72,73,74,75,76].

## 5. Protein *S*-Thiolation/Dethiolation in Protection and Signaling

A major role for GRXs and GRX analogues is the *S*-thiolation/dethiolation of proteins [77,78]. *S*-thiolation is the formation of a mixed disulfide of cysteine with a LMW thiol, such as GSH (*S*-glutathionylation), BSH (*S*-bacillithiolation), or MSH (*S*-mycothiolation). *S*-thiolation has emerged as a major post-translational modification (PTM) of protein cysteines. This PTM protects protein cysteines against permanent damage during oxidative stress when the reactive oxygen species oxidize the thiols (R-SH) to reversible sulfenic acid (R–SOH) and further to irreversible sulfinic (R–SO_2_H) and sulfonic (R-SO_3_H) acid. The irreversible oxidation of –SH causes damage to protein function and LMW thiols like GSH prevent this by reacting with protein disulfides or the sulfenic acid arising from oxidative stress to form mixed disulfides. *S*-thiolation/dethiolation with its fast turnover rate can also act as a redox signal to turn on and off transcription of genes (see Section 3). Once the oxidative stress is removed, either GSH or other LMW thiol or GRXs and their LMW analogues catalyze the reduction of the mixed disulfide and remove the LMW thiol to restore the protein cysteine. *S*-thiolation has not been reported in archaea. However, since all aerobic organisms encounter oxidative stress, *S*-thiolation/dethiolation is expected to occur in archaea.

## 6. Thiol-Dependent Glyoxalases

An important function of LMW thiols that may predate protection against oxidative stress is the detoxification of xenobiotics and endogenously produced electrophiles. The LMW thiol nucleophilic attack on electrophiles can be performed chemically or catalyzed by enzymes. GSH and presumably other thiols react with aldehydes spontaneously to form a hemithioacetal adduct between GSH and 2-oxoaldehydes formed during carbon metabolism [45]. For example, during glycolysis glucose is split into 3-C triosephosphates, glyceraldehyde phosphate and dihydoxyacetone phosphate, which can isomerize to form methylglyoxal, an aldehyde of pyruvate (2-oxoaldehyde) that is a strong electrophile and thus cytotoxic. Methylglyoxal synthase further catalyzes the production of methylglyoxal from dihydroxyacetone phosphate. Methylglyoxal reacts with GSH to form *S*-lactoylglutathione, a reaction catalyzed by glyoxalase I (glyoxylase I). Glyoxalase II (glyoxylase II) hydrolyses *S*-lactoylglutathione to d-lactate and GSH.

While archaea use alternative pathways for central metabolism (e.g., methanogenesis, modified Entner-Doudoroff or modified Emden-Meyerhof-Parnas) [79,80], these microbes produce methylglyoxal and likely use LMW thiols to remove this intermediate based on the following evidence. Glyoxalase I/II (EC 4.4.1.5/EC 3.1.2.6) homologs are relatively common among archaea and most haloarchaea harbor methylglyoxal synthase (EC 4.2.3.3) homologs. Even, the methanogen *M. jannaschii* that does not harbor a methylglyoxal synthase homolog is still found to produce methylglyoxal through an alternative route as a biosynthetic intermediate of 6-deoxy-5-ketofructose-1-phosphate, a precursor of aromatic amino acids [81]. Oren and Gurevich [82] examined eight species of *Halobacteriaceae* for the presence of methylglyoxal synthase and glyoxalase activity. Glyoxalase activity was detected in all eight species while methylglyoxal synthase activity was detected in six out of the eight species. *Haloferax volcanii* extracts showed optimal glyoxalase activity at pH 7 in the presence of 3 M KCl and GSH or γGC [82] revealing that LMW thiols could be used to remove cytotoxic electrophiles produced during metabolism.

## 7. GSH Dependent Formaldehyde Dehydrogenases

Another well-known reactive aldehyde, formaldehyde, is also endogenously produced as a result of catabolism of methionine, methanol, and glyoxalate, or the oxidative demethylation of DNA and RNA. Formaldehyde is toxic [83] and can react with GSH spontaneously or enzymatically (in some eukaryotes and bacteria by *S*-(hydroxymethyl)glutathione synthase (EC 4.4.1.22)) to produce *S*-hydroxymethylglutathione (*S*-HMGSH) [84,85]. *S*-HMGSH is less toxic than formaldehyde and is recycled through oxidation by *S*-HMGSH dehydrogenase (EC 1.1.1.284) to generate *S*-formylglutathione and NAD(P)H [86]. Based on KEGG classification to EC 1.1.1.284, this enzyme appears common to eukaryotes and bacteria but rare in archaea. S-formylglutathione hydrolases (FGSH, EC 3.1.2.12) subsequently hydrolyze the *S*-formylglutathione to GSH and formate. While diverse in primary sequence, FGSHs have a canonical alpha/beta-hydrolase fold and serine hydrolase catalytic triad as exemplified by *E. coli* FrmB and YeiG and yeast YJL068C (PDB: 1pv1) [87]. Based on InterPro family IPR014186 classification, FGSHs appear more common in eukaryotes and bacteria than archaea, with the 40 archaeal hits primarily from the *Sulfolobaceae* family (Figure 2). Thus, an alternative route is likely used to mediate formaldehyde detoxification in archaea.

## 8. Glutathione *S*-Transferases

A superfamily of enzymes, glutathione *S*-transferases (GST), are involved in detoxification and metabolism of endogenous compounds and defense against oxidative stress [88,89,90]. This GST superfamily consists of numerous enzyme classes with different types of reactions, such as nucleophilic aromatic substitution, nucleophilic addition and substitution, conjugate addition, epoxide ring opening, thiolysis, isomerization, and hydrolytic dehydrogenation. GSTs can also act as disulfide bond reductases, dehydroascorbate reductases, peroxidases, thiocyanate reductases, reductive dehalogenases, and alkylarsenate reductases. Moreover, GSTs can be involved in deglutathionylation of protein cysteines. GSTs are not closely related in sequence similarity but share a structural similarity, with a TRX-like N-terminal domain, which binds GSH and a C-terminal domain, which binds the substrate. Mashiyama and colleagues [91] constructed a network based on structural and sequence similarity from 13,000 cytoplasmic GST sequences, including archaeal sequences [91]. Sixteen proteins from the haloarchaea (*Halobacteria* class) fell into the subgroup Xi.1, which consists of GSTs with *S*-glutathionyl-*p*-hydroquinone reductase activity [91]. Interestingly, these enzymes can catalyze GSH-dependent thioltransferase reactions and deglutathionylation reactions along with reduction of GS-hydroquinones to hydroquinones [92]. An InterPro search also yielded 381 archaeal homologs that clustered to the GST-Omega family, IPR016639, which contains glutathionyl-*p*-hydroquinone reductase (Figure 2). Oztetik and Cakir (2013) [93] demonstrated that *Haloarcula hispanica* contained both GSH and had GST activity with 1-chloro-2,4-dinitrobenzene (CDNB), a classic GST substrate. Analysis of the *H. hispanica* genome reveals the presence of *gshA*, at least three GSTs, and GRX homologs. No GshB is apparent. As the GSH concentration was determined using Ellman’s reagent ((5,5′-dithiobis-(2-nitrobenzoic acid) or DTNB)), which would oxidize all –SH groups, including that of γGC, it is likely that the LMW thiol that was measured was γGC and not GSH. Recently, the purification, crystallization, and structure determination of NmGHR, a GST of the Xi class from the extreme haloalkaliphilic archaeon *Natrialba magadii* ATCC 43099 was reported [94]. NmGHR activity was examined using γGC and GSH as co-substrates with: (i) benzoquinone as a substrate for glutathionyl-*p*-hydroquinone reductase activity, (ii) CDNB and ethacrynic acid as substrates for GST activity, and (iii) docosahexaenoic acid (DHA) and bis(2-hydroxyethyl) disulfide (HED) as substrates for thioltransferase activity. No activity was detected for any of these substrates; whether this is due to the method used to purify the enzyme (e.g., His tag, *E. coli* host) or the choice of substrates remains to be determined [94].

## 9. GSH Metabolism

Gamma-glutamyltranspeptidases (GGTs; EC 2.3. 2.2) catalyze the transfer of γ-glutamyl functional groups from GSH to an acceptor that may be an amino acid, a peptide or water (forming glutamate). GGTs play a key role in degradation of GSH and 224 members of the GGT InterPro family (IPR000101) classify to archaea. The *S. solfataricus* GGT homolog SSO_3216 increases in abundance following oxidative challenge [95]. Heinemann et al. (2014) further reported that *S. solfataricus* GGT reacts with GSH, although primarily in the oxidized form [95]. A closer look at the sample preparation indicated that the *S. solfataricus* cells were lysed with a combination of freeze/thaw cycles, and protein samples were prepared from the resulting cell suspension. For metabolite analysis, cell pellets were resuspended in methanol, chloroform was added, and the samples were shaken for 2 h at 0 °C. In both cases, the LMW thiol including protein cysteine and GSH are likely to oxidize. The low levels of GSH detected could have also been contamination from the media since the media for growth, DSMZ, contains yeast extract. *Ignicoccus hospitalis* gave similar results. Neither *I. hospitalis* nor *S. solfataricus* have homologs that classify to GshA, GshB or GshAB (GshF) InterPro families, although *S. solfataricus* SSO_2815 has low, but significant sequence similarity (43% similarity, 23% identity) with the *M. stadtmanae* GshA [95].

## 10. Thiol/Disulfide Switches in Archaea

Disulfide bond formation between cysteine residues and reduction of the disulfide bond acting as an on-off redox switch for transcriptional regulation is demonstrated in bacteria and eukaryotes [96,97]. In bacteria, the transcriptional regulator OxyR is involved in oxidative stress protection [98]. Oxidative stress causes disulfide bonds to form in OxyR, which changes its oligomerization and binding affinity [99]. Oxidized OxyR binds more readily to various promoters, and activates antioxidant genes [99]. The disulfides in OxyR are preferentially reduced by GRX 1 in vivo although TRX is also able to perform the same function in *E. coli* [100]. In OhrR, another transcriptional factor involved in oxidative stress protection, a key cysteine residue becomes *S*-bacillithiolated upon oxidative stress [101], i.e., the LMW thiol, BSH, forms a disulfide bond with the cysteine. The *S*-bacillithiolated OhrR repressor is inactive, which leads to the induction of expression of the gene encoding OhrA, a peroxiredoxin that can detoxify lipid peroxides [102]. The removal of the BSH is performed by bacilliredoxins, analogues of GRXs [78]. A redox switch can also be a part of bacterial two component systems (TCS); for example, the kinase RegB from *Rhodobacter capsulatus* is inactivated by disulfide bond formation under oxidizing conditions [103].

In the archaeal order *Thermococcales*, the sulfur response regulator, SurR, has a thiol-disulfide redox switch that allows *Thermococcales* such as *Pyrococcus furiosus* to change between two different metabolic modes. *P. furiosus* produces hydrogen gas in the absence of elemental sulfur (S^0^) and H_2_S in the presence of S^0^ [104,105]. SurR contains a CXXC motif that functions as a redox-active switch that controls its DNA binding affinity [106]. Oxidation of cysteines with S^0^ inhibits DNA binding by SurR, leading to deactivation of genes related to H_2_ production and derepression of genes involved in S^0^ metabolism [107]. The oxidation can be reversed by addition of excess dithiothreitol (DTT), a reducing agent. Lim et al. (2017) demonstrated that the in vivo reductant is a protein disulfide oxidoreductase in *Thermococcus onnurineus* NA1 [108]. Interestingly, two of the three TRXs that were tested did not reduce SurR [108] suggesting protein substrate specificity within the TRX systems.

Another redox sensitive transcriptional regulator, MsvR, has been described in the strict methanogenic anaerobes, *Methanosarcina acetivorans* and *Methanothermobacter thermautotrophicus*. MsvR displays differential DNA binding under oxidizing and reducing conditions in both of these archaea [109,110]. In *M. acetivorans*, treatment of MsvR with H_2_O_2_ results in oxidation of cysteine thiols, preventing binding of MsvR to promoters. Incubation of oxidized MsvR with the *M. acetivorans* TRX system, consisting of NADPH, TRX reductase and one of the 7 TRXs, leads to reduction of the cysteines and binding to its own promoter [111].

Redox signaling also appears to occur through a thiol-dependent phosphorylation cascade in methanogens. Feige and Frankenberg-Dinkel [112] find that RdmS, a tyrosine kinase with a heme cofactor, undergoes redox-dependent autophosphorylation in *M. acetivorans*. The heme cofactor does not affect RdmS autophosphorylation activity, and the autophosphorylation only occurs under oxidizing conditions [112]. An intramolecular disulfide bond is present in RdmS under oxidizing conditions, and incubation with DTT or CoM abrogates the autophosphorylation revealing the importance of disulfide bond(s) in this mechanism. The authors proposed that either the TRX/TRX reductase system or methanoredoxin may be the natural reductant for the disulfides.

## 11. Coenzyme A in Archaea

CoA has been suggested to be a major thiol in archaea, possibly due to the presence of disulfide reductases (CoADR) that reduce oxidized CoA denoting the ability of these cells to recycle CoA to its reduced form [113,114,115]. CoADR classify to the InterPro IPR017758 family and include 60 hits to hyperthermophilic archaea of the orders *Sulfolobales* and *Thermococcales* (Figure 2). In the hyperthermophilic archaea, Hummel et al. (2005) [116] measured CoA levels in units of µmol/g dry weight at 1.54 in *Thermococcus litoralis,* 0.98 in *P. furiosus,* and 0.4 in *S. solfataricus*. In addition, these authors reported that growth on sulfur increased the CoA levels in these organisms [116]. The CoADRs from *P. horikoshii* and *P.* furiosus are able to use both NADPH and NADH as substrates, unlike the mesophilic bacterial CoADR of *Staphylococcus aureus* [117]. However, several questions remain as to how CoA could assume the protective functions of a LMW thiol. CoA cannot be used as a storage form of cysteine, since the Cys moiety is decarboxylated during CoA biosynthesis. Furthermore, while an unusual NADPH-dependent disulfide reductase with a high affinity for CoA disulfide, which would maintain CoA in a reduced condition, is present in *S. aureus* [113] and *B. anthracis* [118], it is absent in other related species, such as *B. subtilis.* Finally, the pKa of the CoA thiol is very basic (pH = 10) indicating that it is unable to participate in oxidation and reduction reactions. The arguments for CoA serving as the major LMW thiol are: i) It is more resistant to auto-oxidation than cysteine and GSH and ii) it is more stable than GSH in the presence of copper, even at high temperatures [113].

## 12. MSH in Archaea

In high GC actinobacteria, MSH is the major thiol. MSH has an acetylated cysteine with the amino group forming an amide bond with glucosamine, which is linked to inositol (Figure 1) [16,27,119,120]. MSH biosynthesis proceeds through a five-step pathway [119]. The initial substrates, 1L-*myo*-inositol-1-monophosphate and UDP-*N*-acetylglucosamine, react to form *N*-acetylglucosaminylinositol phosphate; this reaction is catalyzed by the *N*-acetylglucosamine transferase, MshA [121,122]. An unidentified phosphatase dephosphorylates this molecule to yield N-acetylglucosaminylinositol, which is deacetylated by an MshB deacetylase [123]. The resulting glucosaminylinositol is ligated with L-cysteine in a reaction catalyzed by a ligase, MshC (IPR017812) [124,125]. The cysteinylglucosaminylinositol is then acetylated to form MSH in a reaction catalyzed by MshD acetyltransferase (IPR017813) [126]. Mca, mycothiol conjugate amidase (IPR017811), catalyzes the cleavage of the amide bond between an electrophile and glucosamine and, thus, plays a major role in MSH dependent detoxification and recycling of MSH [127,128]. Mtr, mycothione reductase (IPR017817), catalyzes the NADPH dependent reduction of oxidized MSH [129]. InterPro analysis of these MSH-related gene homologs reveals only a single hit for MshD (IPR017813) in uncultivated samples. MSH has also not been detected in archaea. This absence of MSH and absence of gene homologs involved in MSH metabolism indicates clearly that MSH is not present in archaea thus far sequenced.

## 13. BSH in Archaea

In low GC Gram positive bacteria, such as the Firmicutes, BSH (Figure 1) is the major LMW thiol. BSH is structurally similar to MSH in that it contains the core cysteinylglucosamine moiety [130]. However, BSH does not contain the N-acetyl residue at the cysteine and the cysteinylglucosamine is linked to l-malate instead of the inositol. Because BSH shares the common cysteinylglucosamine moiety as MSH, the BSH biosynthesis pathway shares a common biosynthetic process to MSH. The first enzymatic reaction was identified as consisting of the joining of UDP-*N*-acetylglucosamine to l-malate catalyzed by a glycosyltransferase (BshA) to yield *N*-acetylglucosaminylmalate (GlcNAcMal) [131]. Next, a deacetylase (BshB) deacetylates GlcNAc-Mal to yield glucosaminylmalate (GlcN-Mal) [132]. The third enzymatic reaction was proposed to involve the ligation of l-cysteine to GlcNAc-Mal to form BSH, a reaction catalyzed by a BSH synthetase (BshC) [133,134]. InterPro search of BshA (IPR023881) and BshC (IPR011199) homologs in archaea resulted in 39 hits for each enzyme, with the majority of hits clustering to the Asgard (an archaeal superphylum with close ties to the last common eukaryotic ancestor [135]). A search for the recently identified BSH disulfide reductase, Ypd, resulted in only two gene homologs in archaea (IPR023856) [136,137]. BSH has not been detected in archaea but Asgard archaea have only recently been cultivated [138], and the presence of BshA and BshC homologs suggests that archaea may be capable of synthesizing BSH.

## 14. Aerobic and Anaerobic Biosynthesis of EGT in Archaea

Another LMW thiol that is common in bacteria is EGT (Figure 1). EGT is a thiourea derivative of histidine, containing a sulfur atom on the imidazole ring, that exists as a thione under physiological conditions [139]. Synthesis of EGT was first elucidated in mycobacteria where the biosynthetic genes are present in a five gene cluster (*egtABCDE*) [11]. First, similar to GSH synthesis, synthesis of γGC is catalyzed by EgtA. Hercynine is formed by the methylation of l-histidine, a reaction catalyzed by a methyltransferase (EgtD). EgtB then catalyzes the addition of γGC to hercynine to form hercynal γGC sulfoxide. Glutamate is removed by a glutamine amidotransferase (EgtC) to form hercynlcysteine sulfoxide. Finally, a pyridoxal 5-phosphate-dependent β-lyase (EgtE) converts hercynlcysteine to EGT. Phylogenetic studies of fungal and bacterial species reveal that synthesis of EGT might not require all five steps. For example, cyanobacteria produce high levels of ergothioneine without orthologs to *egtC*, or *egtE* [140]. In fact, a survey of over 2,500 bacteria showed that the five gene cluster was specific to Actinobacteria and only EgtB and EgtD are the key enzymes for EGT synthesis. The functions of EgtA, EgtC, and EgtE are potentially performed by other unknown enzymes or are not needed since the source of sulfur differs from γGC to cysteine [141]. EgtB (IPR017806) and EgtD (IPR035094) homologs are present in archaea (104 for EgtB and 118 for EgtD) suggesting that EGT may be synthesized in some archaeal species.

Recently, Seebeck and colleagues reported the anaerobic synthesis of EGT [142,143]. The green sulfur bacterium *Chlorobium limicola* encodes a sulfur transferase EanB (Clim_1149, PDB: 6H9A) that converts trimethylhistidine into EGT using oxygen-independent chemistry. The rhodanese-like enzyme transfers sulfur to a non-activated carbon of the trimethylhistidine [143]. The EanA methyltransferase (Clim_1148) converts histidine to trimethylhistidine to initiate this pathway. In archaea, homologs of EanA and EanB are present particularly in anaerobic methanogens. *Methanococcoides vulcani* SAMN04488587_0183 contains an EanA like domain and SAMN04488587_0184 contains a rhodanese domain reminiscent of a sulfur transferase EanB; thus, genome synteny providing further support for this relationship to EGT biosynthesis. The presence of the EGT biosynthesis genes (aerobic and anaerobic) suggests that EGT may play a role in archaea even though this LMW thiol has not been detected in archaea.

## 15. Conclusions and Future Directions

Herein, we show that: (i) γGC is likely the major thiol in haloarchaea and is likely present in methanogens, (ii) CoA is the major thiol in hyperthermophiles, (iii) BSH may be present in archaea due to the presence of BSH biosynthesis genes, and (iv) similarly, EGT may also be present in archaea as both aerobic- and anaerobic-type EGT biosynthesis genes are detected among the archaeal genomic sequences. In addition, novel thiols with unique structures and functions are likely to also be present in archaea. These LMW thiols may not be easily discovered by interrogating just genomic sequences but will require purification and structural characterization of the compound. A better understanding of the structure and function of LMW thiols in archaea will open up novel biochemistries which explain how archaea are able to adapt to extreme environments.

## Figures and Tables

**Figure 1 antioxidants-09-00381-f001:**
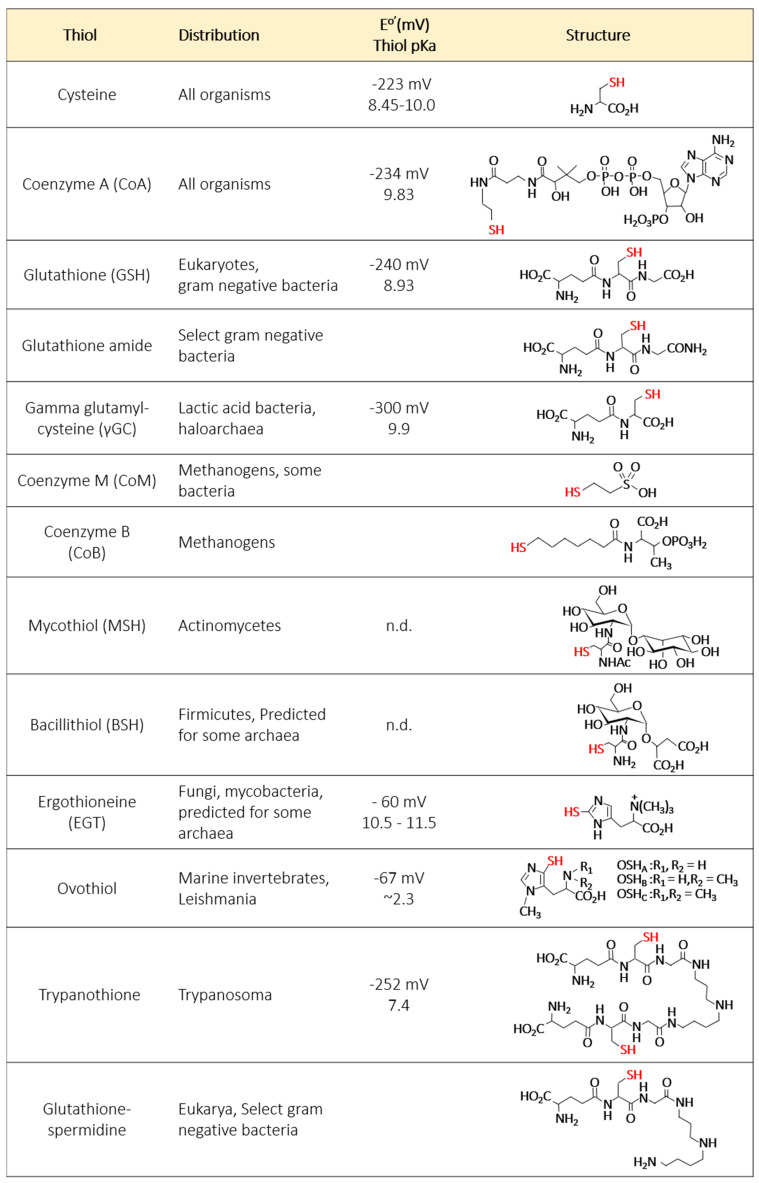
The chemistry of low molecular weight (LMW) thiols and their distribution among phylogenetic groups.

**Figure 2 antioxidants-09-00381-f002:**
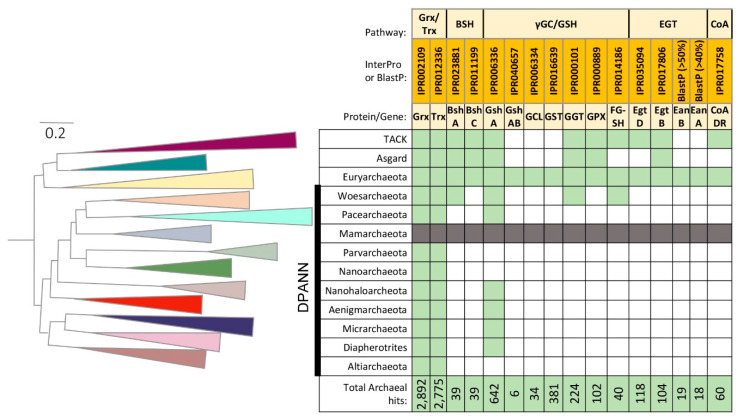
Archaeal protein homologs associated with thiol chemistry and their phylogenetic distribution. Left: Phylogenetic relationship of archaea based on Castelle et al. [39]. Phyla that cluster to DPANN indicated by black vertical bar. Right: Classification of archaeal homologs based on InterPro families and BlastP (See Appendix A for details). Green boxes, archaeal groups with homologs. Grey boxes, Mamarchaeota genome unavailable for analysis.

**Figure 3 antioxidants-09-00381-f003:**
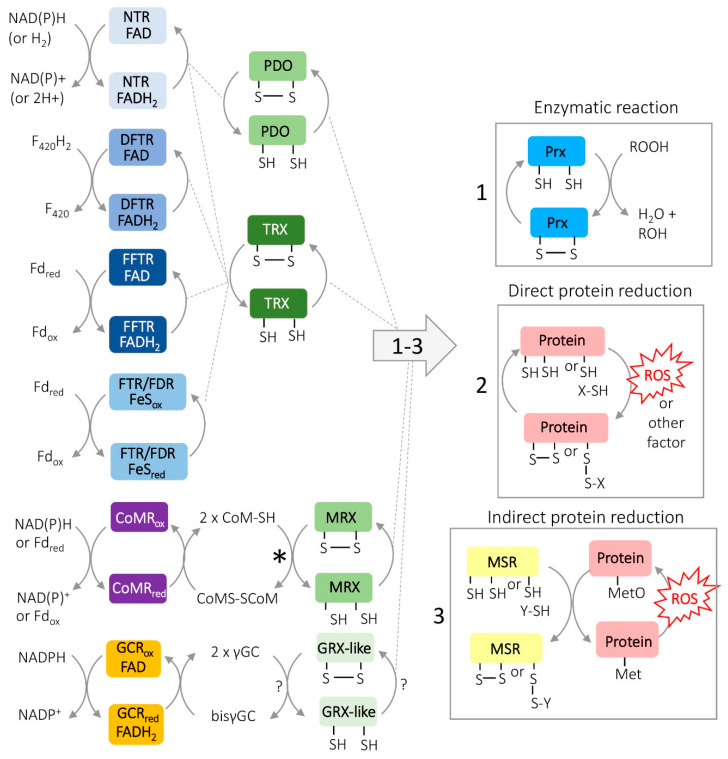
Protein disulfide relay systems. TRX, thioredoxin; GRX-like, glutaredoxin-like; MRX, methanoredoxin (GRX-like); and PDO, protein disulfide oxidoreducatase (GRX-like); NTR, NADPH-dependent TRX reductase; DFTR, deazaflavin (F_420_)-dependent flavin containing TRX reductase; FFTR, ferredoxin-dependent flavin TRX reductase; FTR, ferredoxin: TRX reductase that uses an active site [4Fe-4S] cluster; FDR, ferredoxin: disulfide reductase enzymes that uses an active-site [4Fe–4S] cluster; CoMR, coenzyme M disulfide reductase; CoM-SH and CoM-S-S-CoM, reduced and oxidized forms of coenzyme M; GCR, bisγGC reductase; *, non-enzymatic reduction, ?, not demonstrated. Routes for protein thiol reductant (which may be general or specific): (1) catalytic reactions such as catalyzed by Prx, thiol-dependent preoxiredoxins; (2) direct protein reduction; and (3) indirect protein reduction such as methionine sulfoxide reductase (Msr) catalyzed reduction of methionine sulfoxide (MetO) residues on oxidized proteins; ROOH, alkyl hydroperoxide; ROH, alcohol; ROS, reactive oxygen species or other oxidant.

**Figure 4 antioxidants-09-00381-f004:**
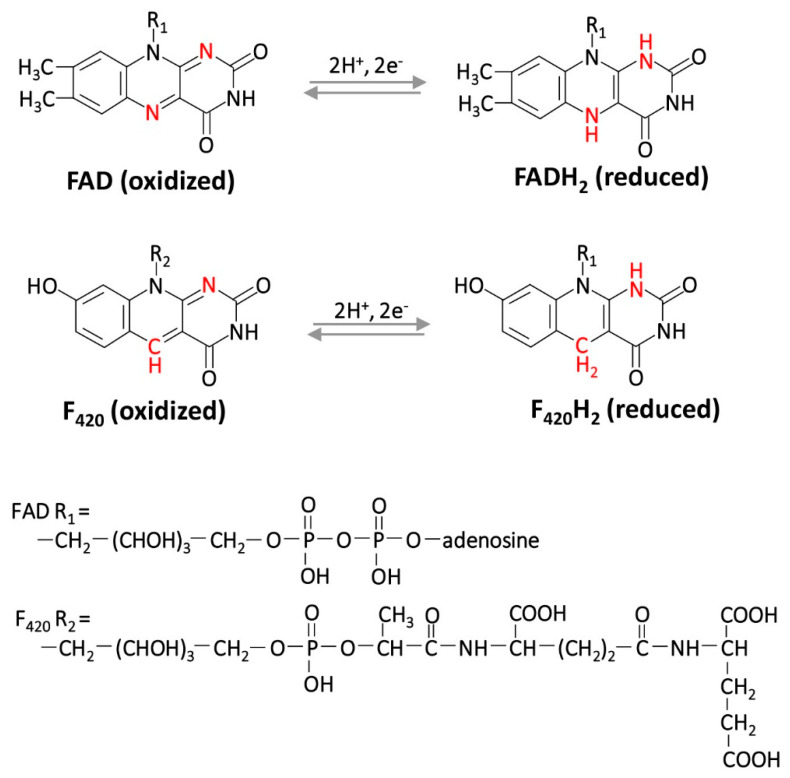
Comparison of coenzyme F_420_ and FAD oxidized and reduced forms.

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
