# Peer review of "Redox and Thiols in Archaea"

_antioxidants, 2020, doi:10.3390/antiox9050381_

Round 1

Reviewer 1 Report

This is a valuable review that brings together a range of information concerning the biochemistry of thiols and their distribution in prokaryotes, with particular emphasis on Archaea. The review is well organised and written, except for some very minor edits as suggested below

  • Be consistent with gamma-glutamylcysteine abbreviation throughout
  • p.11, l. 352 unclear sentence
  • p.12, l. 364 add UDP in Appendix A
  • p.12, l. 391 closing round bracket missing

Author Response

Comments and Suggestions for Authors

This is a valuable review that brings together a range of information concerning the biochemistry of thiols and their distribution in prokaryotes, with particular emphasis on Archaea. The review is well organised and written, except for some very minor edits as suggested below

    Be consistent with gamma-glutamylcysteine abbreviation throughout

Response: We now use γGC consistently throughout the manuscript. We no longer alternate with γ-GC.

    p.11, l. 352 unclear sentence

Response: We corrected the sentence with stating cysteine instead of CoA. This correction now clarifies the sentence.

    p.12, l. 364 add UDP in Appendix A

Response: UDP is now defined in Appendix A.

    p.12, l. 391 closing round bracket missing

Response: The closing round bracket is now added.

Reviewer 2 Report

Manuscript antioxidants-775988 by Rawat and Maupin-Furlow

For the domains of Eukarya and Bacteria various studies of low molecular weight (LMW) thiols have been published in the past, leading to a better understanding of the role and biochemistry of LMW thiols in these organism groups. Therefore, it is known that LMW thiols play important roles such as regulating the redox homeostasis inside cells or as cofactors. However, for Archaea not much is known about the role and presence of LMW thiols until now. This manuscript gives an overview on the state of art of LMW thiols in Archaea. The authors summarize the distribution of currently known LMW thiol classes in archaea and the role and biochemistry of single molecules. The article is well structured and supported by well-designed figures.

The only issue I have in terms of content is that I missed somewhat a small paragraph on thioredoxins in Archaea. Specifically for the methanogens, some literature is available. The authors could easily integrate it into paragraph 3.2 or split this paragraph.

Specific comments:

  1. 67 Transition between the two sentences is missing/not fluent: ‘…are found in domestic animals or humans [25]. Variations of GSH exist…’; please rewrite
  2. 276-276 to me this sentence sounds like DTNB reduces the -SH groups of present LMW thiols: ‘…using DTNB, which would reduce all -SH groups…’, the -SH groups represent the reduced form, to my knowledge DTNB itself gets reduced whereas the LMW thiols gets oxidized by forming a mixed disulfide; please clarify what is meant here
  3. 311-312 not the protein is induced but the expression of ohrA
  4. 332-333 transition between the two paragraphs is missing/not fluent; please rewrite
  5. 334 the Fiege and Frankenberg-Dinkel article showed that RdmS is not a histidine kinase but a tyrosine kinase
  6. 371-375 sentence difficult to follow and understand, or is there meant to be a period instead of a comma in l. 374 ‘…[121,122], Mtr,…’Mtr; please rewrite

Minor:

Inconsistent usage of ‘Archaea’ and ‘archaea’, not clear to me why sometimes with capital letter and sometimes not, e.g. l.127 ‘Archaea and bacteria’. Might be used to differ between the domain Archaea and archaea as organisms in general.

  1. 30 such as
  2. 38 the canonical pathway
  3. 76 auto-oxidation
  4. 107 Either ‘The InterPro database’ or omit ‘The’
  5. 109 Consistent naming of citations in running text, either ‘&’ or ‘and’ between names of authors
  6. 140 ‘based on’ instead of ‘based in’
  7. 187 is there an ‘as’ missing: ‘can also act as a’?
  8. 202 ‘insulin’ instead of ‘inulin’
  9. 210 InterPro
  10. 266 ‘sequences’
  11. 275 introduction of the abbreviation ‘DTNB’ is missing
  12. 281 introduction of the abbreviations ‘DHA’ and ‘HED’ are missing
  13. 322 introduction of the abbreviation ‘DTT’ is missing
  14. 333 correct citation name
  15. 358 ‘than’ instead of ‘that’
  16. 359 ‘even at’ instead of ‘at even’
  17. 387 ‘to involve to’?
  18. 390-391 ‘an archaeal’
  19. 411 better: ‘some archaeal species’

Fig. 2 grey boxes: shade of gray somewhat darker for better contrast

Fig. 3, legend: please add the explanation of FTR with iron-sulfur cluster

Author Response

For the domains of Eukarya and Bacteria various studies of low molecular weight (LMW) thiols have been published in the past, leading to a better understanding of the role and biochemistry of LMW thiols in these organism groups. Therefore, it is known that LMW thiols play important roles such as regulating the redox homeostasis inside cells or as cofactors. However, for Archaea not much is known about the role and presence of LMW thiols until now. This manuscript gives an overview on the state of art of LMW thiols in Archaea. The authors summarize the distribution of currently known LMW thiol classes in archaea and the role and biochemistry of single molecules. The article is well structured and supported by well-designed figures.

The only issue I have in terms of content is that I missed somewhat a small paragraph on thioredoxins in Archaea. Specifically for the methanogens, some literature is available. The authors could easily integrate it into paragraph 3.2 or split this paragraph.

Response: Thanks, we have altered this section of the manuscript to enhance organization and highlight the insight provided on thioredoxins in Archaea including reference to methanogens.

Specific comments:

    67 Transition between the two sentences is missing/not fluent: ‘…are found in domestic animals or humans [25]. Variations of GSH exist…’; please rewrite

Response: Thanks, we have rewritten these sentences to emphasis the theme of this section is focused on GSH synthesis.

    276-276 to me this sentence sounds like DTNB reduces the -SH groups of present LMW thiols: ‘…using DTNB, which would reduce all -SH groups…’, the -SH groups represent the reduced form, to my knowledge DTNB itself gets reduced whereas the LMW thiols gets oxidized by forming a mixed disulfide; please clarify what is meant here

Response: we now state that DTNB would presumably oxidize all of the –SH groups.

    311-312 not the protein is induced but the expression of ohrA

Response: we now state that it [leads to the induction of expression of the gene encoding OhrA]

    332-333 transition between the two paragraphs is missing/not fluent; please rewrite

Response: transition now added.

    334 the Fiege and Frankenberg-Dinkel article showed that RdmS is not a histidine kinase but a tyrosine kinase

Response: this point is now corrected.

    371-375 sentence difficult to follow and understand, or is there meant to be a period instead of a comma in l. 374 ‘…[121,122], Mtr,…’Mtr; please rewrite

Response: we have reorganized this paragraph to more clearly explain this section of the manuscript.

Minor:

Inconsistent usage of ‘Archaea’ and ‘archaea’, not clear to me why sometimes with capital letter and sometimes not, e.g. l.127 ‘Archaea and bacteria’. Might be used to differ between the domain Archaea and archaea as organisms in general.

Response: we now consistently use archaea in a general manner - since we do not specifically state Archaea domain in the text.

    30 such as

Response: corrected.

    38 the canonical pathway

Response: corrected.

    76 auto-oxidation

Response: corrected to autoxidation.

    107 Either ‘The InterPro database’ or omit ‘The’

Response: corrected.

    109 Consistent naming of citations in running text, either ‘&’ or ‘and’ between names of authors

Response: corrected to all ‘and’.

    140 ‘based on’ instead of ‘based in’

Response: corrected.

    187 is there an ‘as’ missing: ‘can also act as a’?

Response: corrected.

    202 ‘insulin’ instead of ‘inulin’

Response: corrected.

    210 InterPro

Response: corrected all Interpro to InterPro.

    266 ‘sequences’

Response: corrected.

    275 introduction of the abbreviation ‘DTNB’ is missing

Response: Ellman's reagent [(5,5'-dithiobis-(2-nitrobenzoic acid) or DTNB)] is now defined.

    281 introduction of the abbreviations ‘DHA’ and ‘HED’ are missing

Response: Docosahexaenoic acid (DHA) and bis(2‐hydroxyethyl) disulfide (HED) are now defined.

    322 introduction of the abbreviation ‘DTT’ is missing

Response: dithiothreitol (DTT) is now defined.

333 correct citation name

Response: We were unclear what is meant by this point but have added that the information is related to the binding of MsvR to its own promoter to clarify which promoter is under discussion if this is what the reviewer meant.

line 333 is related to the following: Incubation of oxidized MsvR with the M. acetivorans thioredoxin system, consisting of  NADPH, thioredoxin reductase and one of the 7 thioredoxins, leads to reduction of the cysteines and binding to its own promoter [105].

The citation references the work by the following which appears appropriate for the information presented:

Sheehan, R.; McCarver, A.C.; Isom, C.E.; Karr, E.A.; Lessner, D.J. The Methanosarcina acetivorans thioredoxin system activates DNA binding of the redox-sensitive transcriptional regulator MsvR. J Ind Microbiol Biotechnol 2015, 42, 965-969, doi:10.1007/s10295-015-1592-y.

    358 ‘than’ instead of ‘that’

Response: corrected.

    359 ‘even at’ instead of ‘at even’

Response: corrected.

    387 ‘to involve to’?

Response: corrected.

    390-391 ‘an archaeal’

Response: corrected.

    411 better: ‘some archaeal species’

Response: corrected.

Fig. 2 grey boxes: shade of gray somewhat darker for better contrast

Response: the shade of gray is now darker for better contrast.

Fig. 3, legend: please add the explanation of FTR with iron-sulfur cluster

Response: FTR/FDR added to legend with explanation and citation.

Reviewer 3 Report

The review “Redox and thiol in Archaea” by Mamta Rawat and Julie A. Maupin- Furlow provides a clear and extensive panorama on a topic regarding Low molecular weigh (LMW) thiols of which up to now there has not been an updated overview. The detailed analysis of the different LMWs in the different kingdoms of life is well written and deepens biochemical pathways of the different LMWs. In the Archaea the role of LMW is still little known but this review has analyzed the different distribution of LMW thiols in this kingdom and has highlighted their possible function. Therefore, this manuscript could be worthy of publication. However, I have some points that, if addressed, would go a long way towards  reducing my reservations  and making this publication even better.

Pag 1 line 13 (Abstract)

Plesae change ……different LMW thiol, bacillithiol…. in …… different LMW thiols, such as bacillithiol

Page 5, 3.2 paragraph

I would like that the authors  describe and comment the role of  PDO not only in relationship  to Sur but also as part of redox system involved in the regeneration of prx in S. sofataricus . PDO activity should be commented

The authors should include the following papers among references :

- Limauro D, D'Ambrosio K, Langella E, De Simone G, Galdi I, Pedone C, Pedone E, Bartolucci S. Exploring the catalytic mechanism of the first dimeric Bcp: Functional, structural and docking analyses of Bcp4 from Sulfolobus solfataricus. Biochimie. 2010 ;92(10):1435-44. doi: 10.1016/j.biochi.2010.07.006

- D'Ambrosio K, Limauro D, Pedone E, Galdi I, Pedone C, Bartolucci S, De Simone G.Insights into the catalytic mechanism of the Bcp family: functional and structural analysis of Bcp1 from Sulfolobus solfataricus. Proteins. 2009 ;76(4):995-1006. doi: 10.1002/prot.22408.

In these papers a new disulfide redox system that reduces peroxiredoxins in Saccharolobus solfataricus is described.  The canonical system NADPH / Tr / Trx,  generally used to reduce Prxs, in S. solfataricus is replaced by the NADPH / Tr / PDO  to regenerate the bacterioferritin (Bcps) comigratory proteins.

Line 147 change glutaredoxin and thioredoxin to Grx and Trx respectively

Line 149 changes glutaredoxin to Grx

Page 6

Fig 3 and legend

In Fig. 3 NTR could also be connected to Prx by Protein Disulfide Oxidoreductase (PDO) as reported in the previous comment. Also  the legend must be completed inserting PDO of S. solfataricus

Line 163 Please change Halobacterium salinarum to H. salinarum

Line 167 Please change Pyrococcus furiosus to  P. furiosus

Page 8

Line 195: Sulfolobus solfataricus must be changed to Saccharolobus solfataricus, based on the update of the taxonomy within the Sulfolobaceae family

Lane 205: Please change Pyrococcus horikoshii in P. horikoshii

Page 10

Lane 288 Edit S. solataricus in S. solfataricus

Lane 298 Edit S. sulfotaricus in S. solfataricus

Lane 320 Insert the follow reference: Lipscomb, G.L.; , Schut, G.J.; Scott RA, Adams, M.W.W. SurR is a master regulator of the primary electron flow pathways in the order Thermococcales. Mol Microbiol 2017, 104, 869-881, doi: 10.1111/mmi.13668

Page 11

Lane 347 Please Change Pyrococcus furiosus in P. furiosus and Sulfolobus solfataricus in S. solfataricus

Author Response

Comments and Suggestions for Authors

The review “Redox and thiol in Archaea” by Mamta Rawat and Julie A. Maupin- Furlow provides a clear and extensive panorama on a topic regarding Low molecular weigh (LMW) thiols of which up to now there has not been an updated overview. The detailed analysis of the different LMWs in the different kingdoms of life is well written and deepens biochemical pathways of the different LMWs. In the Archaea the role of LMW is still little known but this review has analyzed the different distribution of LMW thiols in this kingdom and has highlighted their possible function. Therefore, this manuscript could be worthy of publication. However, I have some points that, if addressed, would go a long way towards  reducing my reservations  and making this publication even better.

Pag 1 line 13 (Abstract)

Plesae change ……different LMW thiol, bacillithiol…. in …… different LMW thiols, such as bacillithiol

Response: we replaced the , with : to avoid confusion and clarify that bacillithiol is one example of a LMW thiol that differs from GSH.

Page 5, 3.2 paragraph

I would like that the authors  describe and comment the role of  PDO not only in relationship  to Sur but also as part of redox system involved in the regeneration of prx in S. sofataricus . PDO activity should be commented

The authors should include the following papers among references :

- Limauro D, D'Ambrosio K, Langella E, De Simone G, Galdi I, Pedone C, Pedone E, Bartolucci S. Exploring the catalytic mechanism of the first dimeric Bcp: Functional, structural and docking analyses of Bcp4 from Sulfolobus solfataricus. Biochimie. 2010 ;92(10):1435-44. doi: 10.1016/j.biochi.2010.07.006

- D'Ambrosio K, Limauro D, Pedone E, Galdi I, Pedone C, Bartolucci S, De Simone G.Insights into the catalytic mechanism of the Bcp family: functional and structural analysis of Bcp1 from Sulfolobus solfataricus. Proteins. 2009 ;76(4):995-1006. doi: 10.1002/prot.22408.

In these papers a new disulfide redox system that reduces peroxiredoxins in Saccharolobus solfataricus is described.  The canonical system NADPH / Tr / Trx,  generally used to reduce Prxs, in S. solfataricus is replaced by the NADPH / Tr / PDO  to regenerate the bacterioferritin (Bcps) comigratory proteins.

Page 6

Fig 3 and legend

In Fig. 3 NTR could also be connected to Prx by Protein Disulfide Oxidoreductase (PDO) as reported in the previous comment. Also  the legend must be completed inserting PDO of S. solfataricus

Response: Thanks, we now include all of this information in the manuscript text in the section which discusses Prxs and the work performed in archaea. The references listed above and citations are now in the main body of the text.

Line 147 change glutaredoxin and thioredoxin to Grx and Trx respectively

Line 149 changes glutaredoxin to Grx

Response: We now use GRX and TRX throughout the manuscript.

Line 163 Please change Halobacterium salinarum to H. salinarum

Line 167 Please change Pyrococcus furiosus to  P. furiosus

Page 8

Line 195: Sulfolobus solfataricus must be changed to Saccharolobus solfataricus, based on the update of the taxonomy within the Sulfolobaceae family

Lane 205: Please change Pyrococcus horikoshii in P. horikoshii

Page 10

Lane 288 Edit S. solataricus in S. solfataricus

Lane 298 Edit S. sulfotaricus in S. solfataricus

Lane 347 Please Change Pyrococcus furiosus in P. furiosus and Sulfolobus solfataricus in S. solfataricus

Response: We have made all of these taxonomic changes and now use appropriate abbreviations.

Lane 320 Insert the follow reference: Lipscomb, G.L.; , Schut, G.J.; Scott RA, Adams, M.W.W. SurR is a master regulator of the primary electron flow pathways in the order Thermococcales. Mol Microbiol 2017, 104, 869-881, doi: 10.1111/mmi.13668

Response: We have inserted the reference as requested.